# Spark Plasma Sintering of Copper Matrix Composites Reinforced with TiB_2_ Particles

**DOI:** 10.3390/ma13112602

**Published:** 2020-06-07

**Authors:** Massimo Pellizzari, Giulia Cipolloni

**Affiliations:** Department of Industrial Engineering, University of Trento, via Sommarive 9, 38123 Trento, Italy; giugicip@gmail.com

**Keywords:** copper, TiB_2_, mechanical alloying, spark plasma sintering, strength, thermal conductivity

## Abstract

The aim of this study is to fabricate a Cu-0.5wt%TiB_2_ composite by mechanical alloying (MA) and spark plasma sintering (SPS). Increasing the milling time, the powders are subjected firstly to a severe flattening process and then to intense welding, which promotes the refinement of TiB_2_ particles, their uniform dispersion in the metal matrix, and the adhesion between the two constituents. Sintered metal matrix composites (MMC) exhibit density values between 99 and 96%, which are generally decreased by increasing milling time in view of the stronger strain hardening. On the other side, the hardness increases with milling time due to the refinement of TiB_2_ particles and their improved distribution. The hardness of MMC is three times higher (225 HV0.05) than the starting hardness of atomized copper (90 HV0.05). Tensile tests show a loss of ductility, but ultimate tensile strength has been increased from 276 MPa of atomized copper to 489 MPa of MMC milled for 240 min. The thermal conductivity of MMC is comparable to that of atomized copper (300 W/mK), i.e., much higher than that of the commercial Cu-Be alloy (Uddeholm Moldmax HH, 106 W/mK) typically used for tooling applications.

## 1. Introduction

The need to improve the properties and performance of components working under severe service conditions has always been the driving force toward new and advanced materials. Moreover, in recent years, the need to ensure different properties in a single material has led the scientific research to focus attention on the production of composite materials. For example, the world of tooling is demanding materials combining high wear resistance and thermal conductivity [1]. Copper, characterized by an intrinsic high electrical and thermal conductivity, is a suitable candidate for new and challenging applications as high-performance materials in thermal and electric fields. However, copper shows a low hardness and poor resistance to wear and fatigue. Therefore, for many applications, it is important to improve the copper properties to match these requirements. An effective method is the production of copper-matrix composite materials reinforced by the dispersion of a hard second phase [2,3].

Mechanical alloying (MA) has been proven to be a good method to produce metal matrix composites (MMCs), because it combines an intense refinement of starting powders and a homogeneous dispersion of the reinforcement. The MMC design is quite elaborate, due to the complexity and to the high number of process variables [4,5]. The goal is to combine the different properties of the two or more constituents in a new material with novel and improved properties. The challenge is to increase the bonding strength of the interphase by controlling the manufacturing process and some particle features, such as size, morphology, and volume fraction [6,7,8,9]. Compared to some liquid phase processes, MA has several advantages; for example, it allows better control of the reinforcement distribution and the production of a more uniform matrix microstructure, reducing segregations [3,10]. Moreover, MA assures a stronger mechanical bonding among the constituents, due to the intense energy involved by impact events between the milling media and the powder. Finally, MA generally involves a lower processing temperature than the liquid phase process, thus reducing the reactivity between the metal matrix and the reinforcement phase. It must be considered that process variables such as the size of the powders, the milling time, the milling speed, and the type of lubricant etc. play an important role during MA [11,12]. Additionally, the type of reinforcement must be properly selected to avoid undesired reactions leading to detrimental phases formation, and to limit the decrease of thermal and electrical conductivity of the copper matrix composite. In previous works, different types of ceramic reinforcements (SiC [13], WC [14,15], NbC [16], Al_2_O_3_ [17], ZrB_2_ [18]) have been used to improve the strength and hardness of Cu, in most cases at the expense of electrical and thermal conductivity. Additionally, TiB_2_ characterized by high hardness (2200 VHN [19]), high elastic modulus (565 GPa), good compatibility with the copper matrix, high thermal stability, and relatively higher electrical (9 × 10^−8^ Ωm [20]) and thermal conductivity (66 W/mK [21]) than other ceramics, has been proposed as a suitable candidate [22,23,24]. Further benefits associated with TiB_2_ dispersion are improved high-temperature creep resistance [25], higher mechanical strength [26,27,28] and wear resistance [21,27,29]

It must be considered that thermal and electrical conductivities are strictly determined by many factors, such as the fractions and the size of the constituents. Experimental and theoretical analyses have revealed that the electrical conductivity of copper matrix composites decreases with increasing particle volume fraction and increases with increasing particle size [30,31]. Indeed, much less thermal conductivity data are available in the literature for Cu-matrix composites, plausibly due to the difficult measurement and the expensive equipment. The presence of porosity is detrimental to physical and mechanical properties, so that the sintering process must be optimized to achieve the maximum possible densification. Among the numerous sintering technologies available nowadays, spark plasma sintering (SPS) is considered one of the most promising looking for full dense material due to the synergic effect of a pulsed electric current and sintering pressure [32,33,34,35]. In view of the lower temperature and the shorter sintering cycle compared to conventional sintering processes, i.e., hot pressing or hot isostatic pressing, SPS provides improved densification with minor microstructural changes and potential chemical reactions [29]. Finally, the ease of operations, the easy control of the energy of sintering, the high reproducibility, and its safety, make SPS a promising and successful sintering technique. Compared to other fabrication methods, a limited number of papers deal with solid-state spark plasma sintered Cu-TiB_2_ composites, starting from mechanically alloyed powders. Just a few of them systematically consider the influence of milling time on the particle size distribution, the reinforcement dispersion inside the Cu matrix and the properties of the MMCs sintered at a relatively low temperature, in the absence of any liquid phase. In [36], hardness up to 2 GPa and relative density close to 92% were reported for Cu-10 wt% TiB_2_ composites. Present authors also showed excellent wear resistance for mechanically milled and SPS’ed Cu + 0.5 wt.%TiB_2_ [37].

The aim of this study is to fabricate a high density Cu-0.5wt%TiB_2_ composite, for tooling applications, by MA and SPS. Firstly, the study is focused on the characterization of the morphology, the particle size distribution, and the TiB_2_ dispersion inside the Cu matrix, as a function of milling time. In the second part, the effects of the MA on the SPS process and the final properties of the composite materials have been evaluated.

## 2. Materials and Methods

A commercial water atomized copper powder (AT-Cu, Pometon S.p.A. Maerne, Italy) with nominal particle size lower than 75 µm was selected to produce MMCs reinforced with 0.5 wt%TiB_2_ (Sigma Aldrich) with particle size less than 3 µm.

The MA process was carried out in a Fritsch Pulverisette 6 planetary mono mill (Idar-Oberstein, Germany) at 400 rpm. Nine powder batches were prepared using milling times comprised between 5 and 240 min, conducted continuously. Each milling cycle has been carried out using a new powder batch to keep a constant filling ratio. Vial and spheres of 10 mm diameter (100Cr6, 63 HRC) were used to obtain a ball to powder ratio (BPR) of 10:1. The system was evacuated down to 1.33 mbar to prevent any possible oxidation during milling. As lubricant, 0.5 wt% of stearic acid (CH_3_(CH_2_)_16_COOH) was used.

The milled powder was cold mounted in epoxy resin for standard metallographic preparation with grinding papers (up to 4000 grit), and final polishing with 3 μm, and 1 μm diamond pastes. The powder morphology was analyzed by optical (OM) and scanning electron microscopy (SEM, model Philips XL30, Hillsborough, OR, USA). The particle size distribution was determined by sieve analysis on 50g powder samples. Each powder batch was poured on the sieves stack (355 to 25 μm, 355, 250, 180, 125, 90, 45, 25 μm), mechanically shaken by a FRITSCH 3 Spartan4 model (Idar-Oberstein, Germany) for 30 min, and finally weighed by a precision balance (0.0001 g). The particle size distribution was obtained by normalizing the sieved mass over the total mass of sample.

SPS was carried using a DR. SINTER SPS 1050 apparatus (Sumitomo Coal Mining Co. Ltd., Tokyo, Japan) mounting graphite punches and die. Samples were heated under a vacuum at 100 °C/min up to 900 °C and subsequently at 50 °C/min up to the sintering temperature of 950°C. The sintering time was 5 min. The temperature was measured by a K-thermocouple inserted into the die, not in contact with the powder sample. Therefore, the temperature value did not correspond to the actual temperature of the sample. A pressure of 60 MPa was applied once the temperature reached 700 °C. The surface C contamination caused by the use of graphite dies has been mechanically removed by grinding in all samples.

Microstructural characterization by OM and SEM was carried after chemical etching with 120 mL of distilled water, 30 mL of hydrochloric acid, and 10 g of iron chloride [38]. Density measurements were carried out using the Archimedes’ principle, according to ASTM B962–08. The relative density of MMCs was calculated following the linear rule of mixture, considering the theoretical densities of Cu as 8.96 g/cm^3^ and of TiB_2_ as 7.76 g/cm^3^. Vickers microhardness was measured according to ASTM E9–08, using an applied load of 0.5 N, a holding time of 10 s, and a loading rate of 0.05 N/s. Measurements were taken in 8 different positions and the average value and standard deviation were reported. Tensile test specimens of 4.9 × 4.9 mm^2^ cross section and a gauge length of 20 mm length were produced directly by SPS. Tests were carried out at a strain rate of 0.1 s^−1^. The fracture surface was analyzed by SEM. Samples for thermal conductivity measurements (disks of D10 mm × 2.7 mm height) were produced by SPS, using the same sintering cycle described above. The thermal conductivity (k) was calculated by the equation k = α∙ρ∙C_p_, where α is thermal diffusivity, ρ is the actual density measured for each sample, and C_p_ is specific heat capacity. The thermal diffusivity and the specific heat were measured using the NETZSCH laser flash apparatus LFA 467 HyperFlash (Selb, Germany). All samples were tested between 400 and 500 °C, with 20 °C temperature steps. The specific heat was determined by the reference method given by ASTM-E 1461‒2011. The LFA was calibrated with a C_p_-standard (Pure-Copper: Ø 10 mm, thickness 2 mm). The thermal conductivity of MMC was compared to that of a commercial Cu-Be alloy (Uddeholm Moldmax HH [39]) typically used for tooling applications. Cu-Be discs were machined from a 40mm diameter bar, delivered in aged condition.

## 3. Results

### 3.1. Powder Characterization: Morphology, Size and Alloying

Initial atomized Cu powder (AT-Cu) and the reinforcing TiB_2_ powder are shown in Figure 1 and their compositions are reported in Table 1. The water-atomized Cu (AT-Cu) shows an equiaxed morphology and typical size in the range of 25–60 μm. The TiB_2_ particles are characterized by an average size of about 3 μm and a hexagonal prism shape.

During milling, the powder mixtures are subjected to high-energy collisions, which result in the occurrence of micro-forging, fracture, and possible agglomeration. The specific morphology observed as a function of milling time (Figure 2) is dependent on the dominant mechanism.

At short milling duration (5–20 min), micro-forging without significant cold welding is evident, leading to a flake-like particle morphology. This is derived from the high plastic deformation of the ductile Cu powder under repeated collisions, caused either by milling medium or the reinforcement. With increasing milling duration (20–120 min), the powders are progressively plastically deformed, and their morphology turns completely into thinner and smaller flakes. After prolonged milling (120–240 min), because of cold welding, the particles show a marked increase in size and their morphology becomes equiaxed. The use of a continuous cycle leads to a drastic increase in the temperature inside the vial, inducing the predominance of welding over fracturing. From Figure 2, it is evident how during MA, the edges of the particle become smoother by increasing milling time, due to the intense compacting action of the milling medium. The cumulative particle size curve and the particle size distributions of powders samples milled for different times are reported in Figure 3.

It should be noticed that the sieving experiments could not always be carried out in the most proper way, owing to a small portion of fine particles sticking on the walls of the sieving apparatus. This explains why for some cumulative curves (Figure 3a), they could not reach 100% of the analyzed powders. However, this factor does not significantly affect the analysis. MA causes a general increase of particle size compared to AT-Cu (dash black line). The mean particle size value (D50) increases by increasing milling time, at a markedly higher rate after 120 min. This can be plausibly explained by the higher temperature achieved by the system during prolonged continuous milling, due to frictional heating [40,41]. After a short milling duration, namely up to 120 min, the powders show a narrow and unimodal distribution (Figure 3b). For longer milling duration (160–240 min), the powders show a shift of the curves towards larger particle size, and a broadening of size distribution, which becomes bimodal (Figure 3b). For a short milling time, the largest particle fraction (45–90 μm), which gives rise to the first distribution peak, shifts towards bigger sizes (90–125 μm). Moreover, for the last three milling times, more than 10% of particles, having sizes bigger than 180 μm, give rise to a second peak, responsible for the bimodal distribution.

According to the dominant MA behaviour, three representative samples have been selected considering the particle size and morphology, namely MMC-5′, MMC-80′, and MMC-240′, and their metallographic cross-sections are shown in Figure 4 to evaluate the distribution of TiB_2_ particles among the Cu powders.

After short milling durations (MMC-5′), TiB_2_ particles are poorly dispersed within Cu particles, as confirmed by a large fraction of TiB_2_ particles mostly located on the Cu surface regions (Figure 4a). During MA, the collisions with the milling media promote the deformation of Cu powders, but modest refinement of TiB_2_ and poor alloying with Cu. Moreover, the adhesion between reinforcement and matrix is not adequate, as demonstrated by the presence of voids between the two constituents (red ring).

When longer milling durations are employed (MMC-80′), the larger plastic deformation led to the formation of layered structures. The MA process proceeds by stacking of flake-like particles which aid the dispersion of TiB_2_ within the matrix [2,40,42]. TiB_2_ looks better dispersed than in MMC-5′, being not only found at the particle surface but also in the subsurface area; by the way, the MA process cannot be considered complete. After 240 min, very few individual TiB_2_ particles which have not been alloyed with Cu are still present, while most hard particles are incorporated within the Cu matrix, giving very homogeneous composite powders (Figure 4c) [42]. Despite the large particle size of copper powder, a refinement of TiB_2_ is evident and this guarantees a more uniform distribution of TiB_2_ and a homogenous microstructure. The good bonding between matrix and reinforcement is now evident; in fact, welding traces almost disappeared and just a few visible voids were detected after 240 min; at this point, MA can be considered complete.

A systematic investigation of chemical analysis evolution during milling was not carried out for present MMCs. Indeed, the authors made it for pure Cu [43], showing that O and C contaminations occur due to stearic acid used as a partition control agent. After 200 min milling, the initial O (0.14 wt%) and C (0.01 wt%) content in AT-Cu (Table 1) increased up to 0.84 wt% and 0.34 wt%, respectively. An intermediate O (0.49 wt%) content is measured after 20 min, while that of C (0.36 wt%) is poorly affected by the milling time. The O and C values decrease to 0.37 and 0.14 wt% during SPS, due to the decomposition of stearic acid.

### 3.2. Spark Plasma Sintering and Microstructures

The densification of MA powders has been evaluated using the displacement of the lower punch of the SPS equipment. The curves of MMC-5′, MMC-80′, and MMC-240′ and their derivate are displayed in Figure 5. To evaluate the effect of MA, the AT-Cu displacement curve is reported as a reference.

Despite the presence of 0.5wt%TiB_2_, the trend of displacement curves during the SPS of mechanically alloyed powders is almost the same as AT-Cu. The sintering process can be divided into two stages: before and after the application of the load at 700 °C. According to a previous study [44] before the application of the load, the densification behaviour of AT-Cu is associable with local deformation and particle rearrangement, which are ruled by powders’ compressibility. The situation is very similar for MMC-5′, where the poor alloying with TiB_2_ does not significantly modify the densification behaviour. However, the presence of hard particles reduces densification with respect to AT-Cu. In the case of MMC-80′, the displacement curve shows gradual and progressive densification already above 200 °C, i.e., at a much lower temperature than AT-Cu and MMC-5′, mostly due to the rearrangement and local deformation of the flake-like particles. However, despite the earlier beginning of the densification of MMC-80′, the softest powder, i.e., AT-Cu, achieves the highest displacement because of the highest densification rate. With an increase in milling time, the presence of hard TiB_2_ particles in mechanically alloyed copper decreases the compressibility of the powder, reducing the densification rate. This effect is even more evident in MMC-240′, where the finer dispersion of the TiB_2_ particle further impairs densification [10]. In the case of MMC-240′, densification proceeds to a very low extent before the application of pressure, suggesting that both re-arrangement and localized deformation do not play a significant role. Rearrangement is very low, in view of the relatively high apparent density of large equiaxed composite powders compared to the flake-like ones [45,46]. Furthermore, a lower deformation is expected, due to the higher strain hardening to which particles are subjected after longer milling time and due to the more efficient dispersion hardening of reinforcement.

The behaviour in the loading regime is rather different. Once the compacting force is applied (700 °C), a fast shrinkage could be observed for AT-Cu, explained by the strong bulk deformation of equiaxed soft metal particles. A slightly lower displacement is shown by MMC-5′, due to the reinforcing TiB_2_ particles. A comparable displacement is shown by MMC-240′, because the loading pressure permits the local deformation, which could not occur under free load conditions. For this reason, the displacement, and the displacement rate curves of MMC-240′, are higher than MMC-80′. On the other hand, very low densification is observed for MMC-80′, since it has already occurred during the load-free regime. A general important observation is that, in all cases, the densification rate falls to zero at about 900 °C, meaning that the maximum densification has occurred.

Some interesting information regarding porosity, reinforcement distribution, and interface bonding between TiB_2_ and the copper matrix is illustrated by optical and SEM micrographs for MMC-5′ (Figure 6a,b).

From Figure 6, it is clear how the microstructures of sintered samples keep the memory of the particle size and morphology of the milled powder [47]. Thus, in MMC-5′ TiB_2_, particles are poorly refined and dispersed inside the matrix (Figure 6a). Moreover, the presence of large TiB_2_ particles is often associated with bad interface adhesion between the reinforcement and Cu (Figure 6b). By increasing milling time, the dispersion of TiB_2_ inside sintered products is gradually improved.

In MMC-80′ (Figure 6c,d), both pores and reinforcement tend to locate at former particle boundary regions. Once the milling process is completed, the powders possessed flake-like morphology, and they keep this shape even after applying the compacting pressure. As a result, upon the completion of sintering, a majority amount of those thin flakes is aligned in the direction perpendicular to that of punch stroke, leading to the distribution of porosity and reinforced particles following the same direction. Nevertheless, large TiB_2_ particles are still present and some longitudinal pores could not be eliminated. The adhesion between the two constituents in MMC-80′ is improved compared to MMC-5′. MMC-240′ (Figure 6e,f) exhibits the best microstructure after sintering: the porosity is very fine and evenly distributed and the TiB_2_ is refined and homogenously distributed.

Visible porosity is still observed, however, it also became much finer in size and dispersed quite homogeneously in the matrix. Another remarkable feature that should be pointed out is the white region located at former particles’ boundaries (Figure 6e), absent in MMC-5′ and MMC-80′. These white areas are portions of the Cu matrix undergoing re-melting or re-crystallization during sintering [37,47], due to the local temperature increase induced by the Joule effect [4,44].

### 3.3. Mechanical Properties

The hardness and relative density values of sintered samples, as a function of milling time, are reported in Figure 7.

The relative density decreases prolonging milling time, due to the strain and dispersion hardening to which powders are exposed. By the way, the relative density of MMCs is acceptable since it is higher than 96% (Figure 7). MA induces an even dispersion of fine hard TiB_2_ fragments into the copper powder, avoiding the formation of agglomerates and assuring the achievement of high relative density. By the way, a residual porosity is still present in the sintered sample. This can also be associated with the incomplete decomposition of stearic acid during SPS. In previous studies [43], it has been demonstrated that the decomposition of stearic acid is controlled by the particle size and morphology of the milled particles. Generally, the decomposition of stearic acid occurs at different temperatures 150, 300 and 450 °C, correlated with different gas emissions: 100–1000 °C, continuous water evaporation; 300 °C, water evaporation and the production of CO and especially CO_2_; 450 °C, H_2_ production [43]. By the way, when particle size increases, the decomposition of PCA is shifted to a higher temperature because stearic acid, due to the intense overlapping of the particles, is constrained between them and its reduction is delayed and limited. The larger and more compacted powders hinder the correct and free decomposition of stearic acid before the application of the SPS load, leading to the decrease of relative density increasing milling time [41,43]. The application of the load at 700 °C hinders any delayed gas release and it is responsible for the lower relative density of MMC-240′ compared to the other MMCs. Since the particle size and morphology of MMC-5′ and MMC-80′ are more suitable for degassing, the decomposition of stearic acid finishes at 600 °C, and the application of the load at 700 °C does not limit the reduction during SPS and it ensures higher values of relative density.

In Figure 7 it can be observed that the hardness is directly proportional to the milling duration, showing an opposite tendency with respect to density. This can be partly explained because of the strain-hardening effect induced by high energy collisions between powder particles and milling media. The longer the milling process, the higher the strain hardening, and therefore the higher the hardness of as-milled powders. This is finally reflected in a higher hardness of the sintered material, since the relatively short sintering time and temperature used in the present work do not allow complete recovery and recrystallization. Generally, when increasing porosity, a reduction of the hardness of the sintered sample is expected, but in this case, the increasing hardness confirms that strain hardening abundantly compensates the negative effect of porosity. Moreover, as shown by microstructural analysis (Figure 6), once prolonging the milling duration, TiB_2_ particles are progressively refined and their distribution becomes progressively more homogeneous. As a result, the hardness trend shows a step increase for milling time longer than 120 min, due to the more homogenous TiB_2_ dispersion (Figure 7).

The most negative, and in some cases, inevitable, drawback of mechanical strain hardening and of particles dispersion by MA in a copper matrix is the decrease of thermal conductivity due to the formation of defects among the matrix and the presence of reinforcement particles acting as a barrier to the thermal flow [48]. In Figure 8, the specific heat and the thermal diffusivity values are reported. Since MMC-5′ does not show any significant improvement in comparison with AT-Cu, it has been neglected for the characterization of the thermal conductivity and tensile test.

As expected, the specific heat increases with increasing the temperature for all the materials following the Debye theory. MMC samples show a higher specific heat than the AT-Cu, especially MMC-240′, due to the more intense strain hardening and the more uniform dispersion of TiB_2_, which limits the Cu free-TiB_2_ conductive areas. The thermal diffusivity slightly decreases with increasing temperature and AT-Cu constantly shows the highest value. Considering the specific heat, the thermal diffusivity, and the density of the samples, it has been possible to calculate thermal conductivity (Figure 9).

The thermal conductivity of MMC-80′ and MMC-240′ is equal to 298 and 297 W/mK, respectively, very similar to the value of AT-Cu (300 W/mK). Despite the mechanical strain hardening and the addition of TiB_2,_ the thermal conductivity of MMCs has not been severely affected. The small amount (0.5 wt%) of TiB_2_, the high intrinsic thermal conductivity of this compound, and its good dispersion, especially for long milling time, lead to an excellent combination of hardness and thermal conductivity in comparison with AT-Cu. These encouraging results open the way to the addition of larger amount of TiB_2_ to further increase the hardness and wear resistance of MMCs. It can be noticed that all the materials exhibit a stable thermal conductivity within the temperature range analyzed (400‒500 °C), which is typical of the service temperature range during the injection molding of plastic materials. Finally, a very important observation is that all the materials produced by MA show a higher thermal conductivity than commercial Cu-Be alloy [39], characterized by a thermal conductivity equal to 106 W/mK.

Figure 10 shows the tensile stress-strain curves of the samples produced by MA.

AT-Cu shows the typical tensile behaviour of ductile materials, exhibiting a continuous strain hardening up to fracture, occurring at about 29% elongation. Its mechanical strength is characterized by a yield strength of 110 MPa and an ultimate tensile strength of 276 MPa (Table 2). MMC-80′ shows an almost perfectly plastic behaviour, consisting of a linear elastic deformation followed by a plastic regime with little strain hardening, and no plastic instability. Its yield strength is more than double (282 MPa) compared to AT-Cu, while its fracture elongation is reduced to about 8%. On one hand, this embrittlement is in line with the hardness increase caused by TiB_2_ dispersion and the strain hardening induced by mechanical milling. On the other hand, the lower ductility of MMC-80′ is also the result of the higher porosity, as confirmed by the lower density (~98%). These effects are further emphasized in MMC-240′, showing a completely brittle behaviour, characterized by linear elastic deformation up to fracture. The ultimate tensile strength is increased to 489 MPa, at the expense of fracture elongation, which drops down to ~1%. Nevertheless, the strong dispersion hardening and the lower density of this material (96%) (Figure 7) play a very detrimental influence on ductility.

The mechanical behaviour just described is reflected in fracture surfaces analyzed by SEM (Figure 11). For AT-Cu (Figure 11a), the fracture morphology reveals typical shallow dimples with different sizes, distributed throughout the fracture surfaces. Dimples are a significant sign of a ductile fracture and, in turn, of the strong bonding which occurs between Cu particles during sintering. The fracture surface seems to evidence a bimodal size distribution of dimples: large (green) and smaller (red) dimples are plausibly associated to the different size of powder particles, undergoing variable deformation/recrystallization during SPS. Another interesting feature is the presence of interparticle cracks, which may highlight some debonding leading to fracture (yellow). The fracture could also initiate within void clusters because of a sequence of voids nucleation, growth, and coalescence (light blue).

For MMC-80′, the fracture surface highlights a partial ductile fracture, still characterized by dimples areas (red circle in Figure 11b). By the way, the elongated voids morphology recalls the flake-like microstructure of the mechanically alloyed powder and is a significant of lack of interparticle bonding: in this material, fracture occurs by interparticle cracks propagation (yellow). A clear change of fracture mode is evidenced for MMC-240′, showing a very brittle fracture morphology (Figure 11c). The brittle nature of the fracture is promoted by the stronger strain hardening and the more effective strengthening effect of TiB_2_. The fracture proceeds by a strong decohesion between the particle, as revealed by the large crack in Figure 11c.

## 4. Conclusions

The microstructure and some important properties of Cu-0.5 wt% TiB_2_ composite fabricated by mechanical alloying and Spark Plasma Sintering in the solid state were investigated. A systematic analysis of the influence of milling time on particle size distribution, and TiB_2_ dispersion in the Cu matrix, allowed us to find the most proper processing parameters and their correlation with mechanical properties and thermal conductivity of sintered MMCs.

For milling times up to 120 min, the predominance of plastic deformation promotes a uniform flake-like powder morphology, accompanied by a slight size increase. For longer milling times, up to 240 min, the occurrence of welding enhances the formation of large equiaxed particles and the uniform dispersion of TiB_2_.

The density of SPSed MMCs generally decreases by increasing milling time, because of the higher strain hardening; by the way, density values are higher than 96%. By increasing milling time, the refinement of TiB_2_ and its more uniform distribution lead to a sharp increase of hardness, from 90 HV0.05 for atomized copper, up to 225 HV0.05 for MMC-240′. The most interesting result, also compared to previous ones reported in literature, is that present MMCs exhibit the same thermal conductivity of copper (300 W/mK), and are thus competitive with the commercial Cu-Be (106 W/mK). Tensile tests show a loss of ductility and a gain in strength for MMC samples. Only MMC-80 partially keeps the ductility of AT-Cu with a fracture surface characterized by dimples. Meanwhile MMC-240′ exhibits a brittle nature of fracture related to the higher hardness, promoted by the stronger strain hardening and the more effective strengthening effect of TiB_2_. The results of the present investigation could be further exploited, considering the effect of milling times comprised between 80 and 240 min on the mechanical properties of MMCs.

## Figures and Tables

**Figure 1 materials-13-02602-f001:**
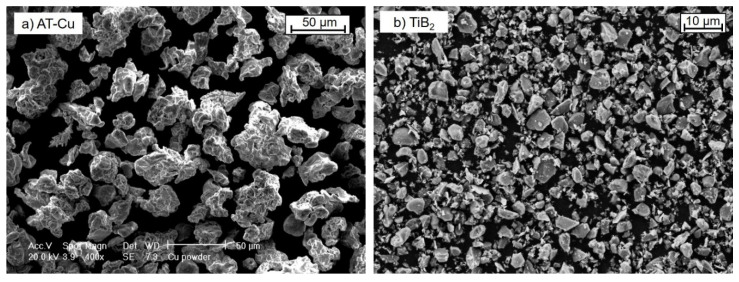
SEM micrographs of (**a**) water atomized Cu powder (AT-Cu) and (**b**) blend of TiB_2_.

**Figure 2 materials-13-02602-f002:**
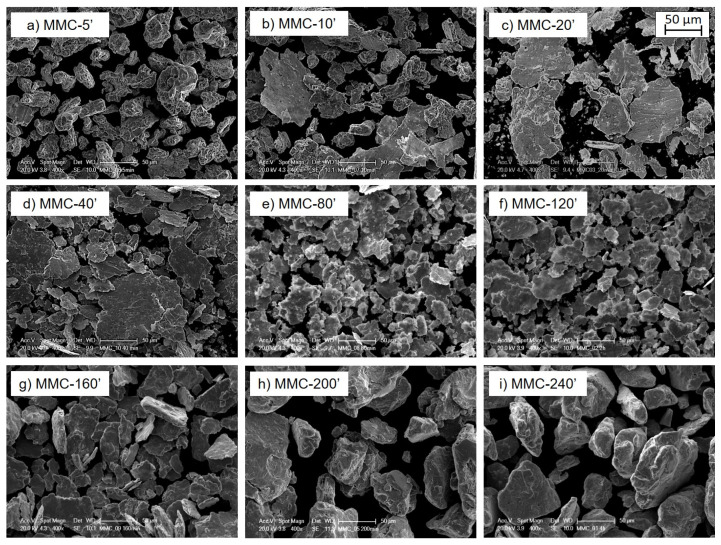
SEM micrograph of MMC powders after different milling times: (**a**) 5′, (**b**) 10′, (**c**) 20′, (**d**) 40′, (**e**) 80′, (**f**) 120′, (**g**) 160′, (**h**) 200′ and (**i**) 240′.

**Figure 3 materials-13-02602-f003:**
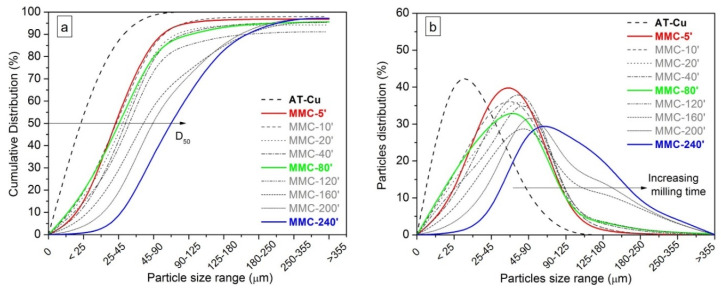
Cumulative distribution (**a**) and particle distribution (**b**) of metal matrix composites (MMC) powder as a function of milling time.

**Figure 4 materials-13-02602-f004:**
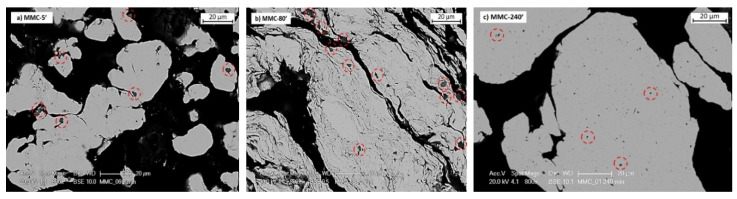
SEM micrographs of powder cross section of (**a**) MMC-5′, (**b**) MMC-80′ and (**c**) MMC-240′.

**Figure 5 materials-13-02602-f005:**
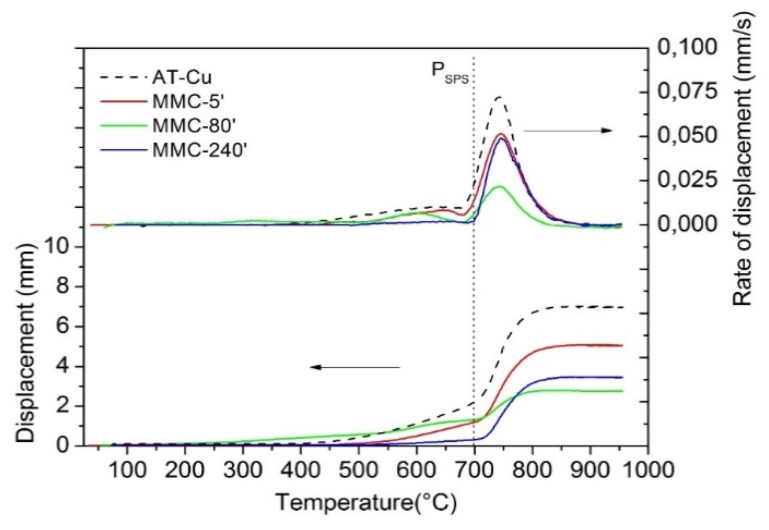
Displacement and displacement rate curves of AT-Cu, MMC-5′, MMC80′, and MMC-240′.

**Figure 6 materials-13-02602-f006:**
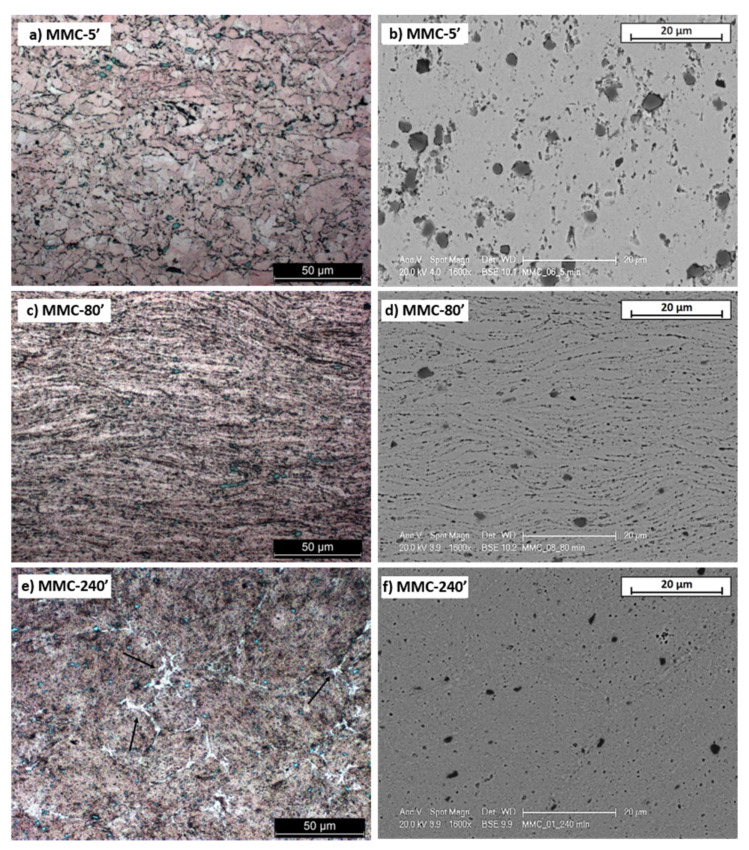
Optical and SEM micrographs of (**a**,**b**) MMC-5′, (**c**,**d**) MMC-80′, and (**e**,**f**) MMC-240′.

**Figure 7 materials-13-02602-f007:**
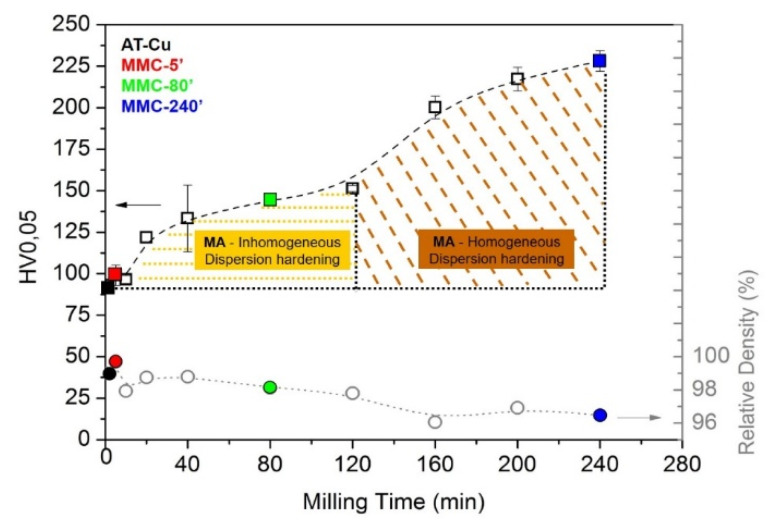
Hardness and relative density of the MMC sample.

**Figure 8 materials-13-02602-f008:**
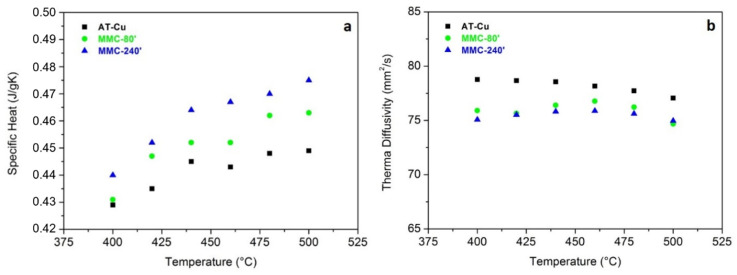
Specific heat (**a**) and thermal diffusivity (**b**) of sintered samples.

**Figure 9 materials-13-02602-f009:**
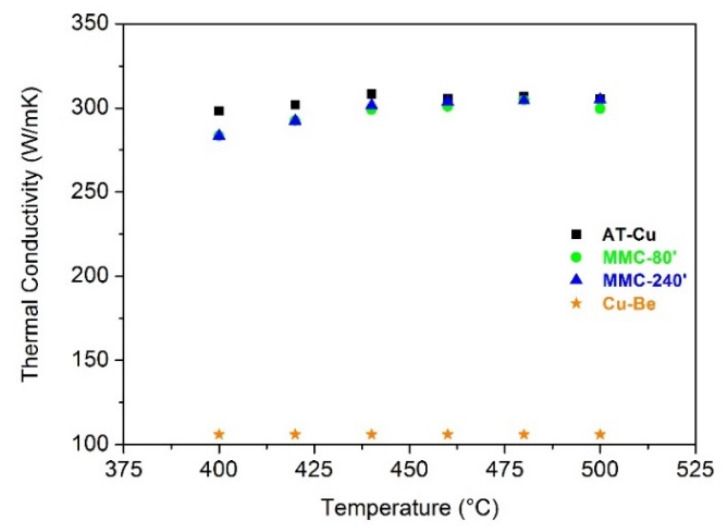
Thermal conductivity of sintered samples.

**Figure 10 materials-13-02602-f010:**
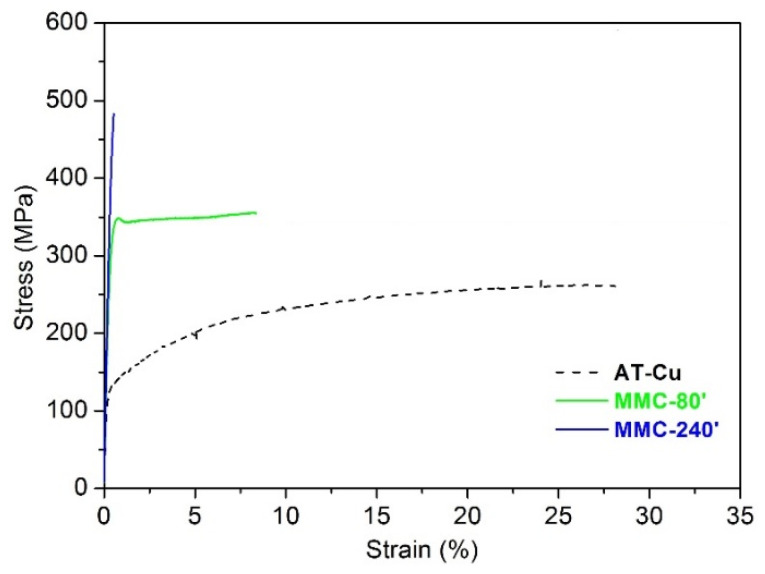
Tensile stress-strain curves of the sintered mechanical alloying (MA) samples and AT-Cu.

**Figure 11 materials-13-02602-f011:**
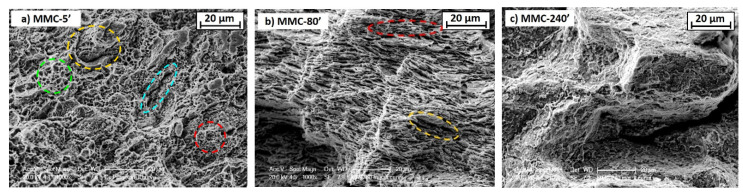
Fracture surfaces (SEM) of sintered samples: (**a**) AT-Cu, (**b**) MMC-80′ and (**c**) MMC-240′.

**Table 1 materials-13-02602-t001:** Nominal composition of the AT-Cu and TiB_2_ powders (weight %).

Powder	Cu (%)	Ti (%)	B (%)	C (%)	O (%)	N (%)	Fe (%)
AT-Cu	99.5	—	—	0.01	0.14	—	—
TiB_2_	—	66	33.2	0.03	0.40	0.15	0.025

**Table 2 materials-13-02602-t002:** Mechanical properties of AT-Cu, MMC-80′ and MMC-240′.

Powder	σ_y_ (MPa)	UTS (MPa)	ε (%)
AT-Cu	110	276	28.8
MMC-80′	282	352	8.2
MMC-240′	451	489	1.2

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
