# Peer review of "Spark Plasma Sintering of Copper Matrix Composites Reinforced with TiB2 Particles"

_materials, 2020, doi:10.3390/ma13112602_

Round 1

Reviewer 1 Report

The main note concerns the introduction. The authors explain the issues taken up, but do not relate to specific results of other authors. Therefore, it is not clear from the article whether anyone did similar tests and what the results were. The authors also do not indicate what new they have achieved compared to their predecessors.

On the other hand, a lot of literature was cited in the introduction, although it is not known what the authors wanted to pay attention to. The general form of citing [12-17], [18-21], [22-24] without providing the essence of this citation is of little use.

On an editorial issue: there should be spaces everywhere between numbers and units.

The way of citing literature is inconsistent - probably different fragments of the text were written by different people. The introduction contains numbers, eg [25]. Later in the article, surnames are given, e.g. [Petzov, 1999].

Author Response

Reply to Reviewer 1

The main note concerns the introduction. The authors explain the issues taken up, but do not relate to specific results of other authors. Therefore, it is not clear from the article whether anyone did similar tests and what the results were. The authors also do not indicate what new they have achieved compared to their predecessors.

The introduction was revised to relate to specific results of other authors, also to emphasize that just a few papers are dealing with solid-state SPS of Cu+TiB2, i.e. at relatively low temperature.

We have also tried to remark the originality of this paper compared to those authored by predecessors, in the Introduction as well as in the Conclusions: in particular, we considered the solid-state sintering of Cu-TiB2 composites, while most of previous results refer to liquid phase sintering. We made a systematic analysis of milling time (0-240 min) on the properties (particle size distribution, TiB2 distribution…) of Cu-TiB2 powders. Finally, we measured the thermal conductivity of MMC’s, showing very promising values compared to pure Cu and commercial Cu-Be alloys.

On the other hand, a lot of literature was cited in the introduction, although it is not known what the authors wanted to pay attention to. The general form of citing [12-17], [18-21], [22-24] without providing the essence of this citation is of little use.

The reviewers’ comment is correct. The introduction was revised to provide the essence of citations.

On an editorial issue: there should be spaces everywhere between numbers and units.

Spaces between numbers and units have been added.

The way of citing literature is inconsistent - probably different fragments of the text were written by different people. The introduction contains numbers, eg [25]. Later in the article, surnames are given, e.g. [Petzov, 1999].

Sorry for this. Citations have been now modified using brackets.

Reviewer 2 Report

The paper titled “Spark Plasma Sintering of Copper Matrix Composites Reinforced with TiB2 Particles” is mostly focused on the MA process of Cu and TiB2 performed at different milling times. The paper is quite interesting but a major revision of the manuscript is still required. The comments are listed below.

  1. The “Materials and Methods” (M&M) section looks inappropriate. Fig. 1, Table 1 and corresponding description should be moved to the beginning of the Results. In the M&M, the text should be more technical and should include a description of the origin of the materials.
  2. The porosity of the obtained samples was analyzed by SEM characterization only. The addition of low-temperature adsorption data is highly advisable.
  3. Figures 6-8 would be better to joint into one.
  4. What is commercial Cu-Be alloy? More information should be provided.
  5. The table in Fig. 12 should be presented separately, and all the parameters should be explained.

Technical remarks:

  1. Check the indexes in formulas and dimensions (TiB2, CH3-(CH2) 16-COOH, g/cm3, mm2, etc.).
  2. Line 14: “which generally decrease” –> “which are generally decreased”?
  3. Line 18: “276MPa … 489MPa” – space is missing. The same is for other units.
  4. Line 31: “thermal conductivity is a suitable candidate” – the comma is missing.
  5. Line 53-55: “In previous works different … of Cu, in most cases at the expense …” – there are missing commas.
  6. Line 131: “At short milling durations (5-20 minutes) micro-forging without” – the comma is missing.
  7. “a flake like particles” -> “the flake-like particles” (lines 132, 173, 200)
  8. Figure 13: Are there SEM images? Please, refine in the figure caption.
  9. Partly, the referring style is inappropriate (lines 106, 153, 174, 179, 205, 209, 228, 249, 250, 251, 262, 267, 272, 293).
  10. Line 155: “duration (160-240 minutes) the powders show” – the comma is missing.
  11. Line 191: “Despite of the presence of 0.5wt%TiB2 the trend” – the comma is missing, "of" should be eliminated.
  12. Line 193: “before and after the application of the load at 700°C (PSPS)” – what is PSPS?
  13. Lines 193-194: “According to previous study [Diouf] before” – the comma is missing, and what is [Diouf]? Incorrect reference?
  14. Line 218: “On the other side very” -> “On the other hand, very”
  15. Line 261: “By the way, a residual porosity in the sintered sample still present.” -> “By the way, a residual porosity is still present in the sintered sample.”
  16. Line 298: “the specific heat increases increasing” -> “the specific heat increases with increasing”
  17. Line 309: “the addition of TiB2 the thermal conductivity” – the comma is missing.
  18. Line 350: “decohesion between the particle as revealed by the large crack in Figure 14-c.” – no Fig. 14 is presented.

Author Response

The authors are very thankful for the reviewers' suggestions and also for the patience in correcting the various typing errors. Revisions are marked in blue in the attached file.

Reply to Reviewer 2

The paper titled “Spark Plasma Sintering of Copper Matrix Composites Reinforced with TiB2 Particles” is mostly focused on the MA process of Cu and TiB2 performed at different milling times. The paper is quite interesting but a major revision of the manuscript is still required. The comments are listed below.

The “Materials and Methods” (M&M) section looks inappropriate. Fig. 1, Table 1 and corresponding description should be moved to the beginning of the Results. In the M&M, the text should be more technical and should include a description of the origin of the materials.

Fig. 1, Table 1 and corresponding description have been moved to the beginning of the Results section. The M&M description was improved using a more technical style. The origin of materials was indicated.

The porosity of the obtained samples was analyzed by SEM characterization only. The addition of low-temperature adsorption data is highly advisable.

The porosity evidenced by SEM is also supported by density values measured by the Archimedes method (Figure 7). These two techniques should give a sufficient description of porosity. Now, this has been also remarked in the description of mechanical tests results.

Figures 6-8 would be better to joint into one.

Done.

What is commercial Cu-Be alloy? More information should be provided.

The commercial Cu-Be ally is Uddeholm Moldmax HH (https://www.uddeholm.com/italy/it/products/moldmax-hh/)

This is now indicated in the Abstract as well as in section 2, Materials and methods.

The table in Fig. 12 should be presented separately, and all the parameters should be explained.

Table and Fig. 12 are now presented separately. The description of mechanical behaviour has been improved, explaining in more detail all the parameters.

Technical remarks:

Check the indexes in formulas and dimensions (TiB2, CH3-(CH2) 16-COOH, g/cm3, mm2, etc.).

Indexes have been corrected.

Line 14: “which generally decrease” –> “which are generally decreased”?

Corrected.

Line 18: “276MPa … 489MPa” – space is missing. The same is for other units.

Spaces between numbers and units have been added.

Line 31: “thermal conductivity is a suitable candidate” – the comma is missing.

Comma has been added.

Line 53-55: “In previous works different … of Cu, in most cases at the expense …” – there are missing commas.

Comma has been added, also in the other parts of the manuscript.

Line 131: “At short milling durations (5-20 minutes) micro-forging without” – the comma is missing.

Comma has been added.

“a flake like particles” -> “the flake-like particles” (lines 132, 173, 200)

Corrected.

Figure 13: Are there SEM images? Please, refine in the figure caption.

Done.

Partly, the referring style is inappropriate (lines 106, 153, 174, 179, 205, 209, 228, 249, 250, 251, 262, 267, 272, 293).

Citations have been corrected.

Line 155: “duration (160-240 minutes) the powders show” – the comma is missing.

Comma has been added.

Line 191: “Despite of the presence of 0.5wt%TiB2 the trend” – the comma is missing, "of" should be eliminated.

Comma has been added, and “of” was eliminated.

Line 193: “before and after the application of the load at 700°C (PSPS)” – what is PSPS?

PSPS has been eliminated.

Lines 193-194: “According to previous study [Diouf] before” – the comma is missing, and what is [Diouf]? Incorrect reference?

Comma has been added, and the citation was corrected.

Line 218: “On the other side very” -> “On the other hand, very”

Corrected.

Line 261: “By the way, a residual porosity in the sintered sample still present.” -> “By the way, a residual porosity is still present in the sintered sample.”

Corrected.

Line 298: “the specific heat increases increasing” -> “the specific heat increases with increasing”

Corrected.

Line 309: “the addition of TiB2 the thermal conductivity” – the comma is missing.

This sentence has been reformulated.

Line 350: “decohesion between the particle as revealed by the large crack in Figure 14-c.” – no Fig. 14 is presented.

Corrected.

Reviewer 3 Report

Dear authors,

You presented a very harmonious and interesting paper. This manuscript will be of interest to as specialists so readers from related fields. Despite the logical presentation of the results, I have questions:

  1. Why in the formula for the thermal conductivity did you give the bulk material density? In fact, you got material with residual porosity.
  2. From the above results, it is not clear to me what happens to the chemical composition after MA and SPS, namely, how does the concentration of oxygen (in the sample body) and carbon in the surface layers change?

Please pay your attention to the writing of superscripts in lines 109, 113 and 114.

         Good luck!

Author Response

The authors are very thankful for the suggested revisions. All issues (marked in blue)  have been addressed in the attached manuscript.

Reviewer 3

Dear authors,

You presented a very harmonious and interesting paper. This manuscript will be of interest to as specialists so readers from related fields. Despite the logical presentation of the results, I have questions:

  1. Why in the formula for the thermal conductivity did you give the bulk material density? In fact, you got material with residual porosity.

In fact, we have used the actual density measured for each sample in the formula. We missed to clarify this in the description of methods. We now added this detail.

  1. From the above results, it is not clear to me what happens to the chemical composition after MA and SPS, namely, how does the concentration of oxygen (in the sample body) and carbon in the surface layers change?

We did not carry out a systematic chemical analysis investigation for present MMC’s. Indeed, we made it for pure Cu [47], showing that there is O and C contamination due to the stearic acid used as partition control agent. After 200 min MM, the initial O (0.14 wt%) and C (0.01 wt%) content in AT-Cu raise up to 0.84 wt% and 0.34 wt%, respectively. Intermediate O content is measured after 20 min milling, while that of C is poorly affected by the milling time. The O and C values decrease to 0.37 and 0.14 wt% during SPS, due to the decomposition of stearic acid. Indeed, the surface C contamination from graphite dies has been mechanically removed by grinding in all samples before measuring density, and other properties.

This is now reported in the text, at the end of paragraph 3.1.

Please pay your attention to the writing of superscripts in lines 109, 113 and 114.

Thank you, we corrected all superscripts and also some more typing errors.

Round 2

Reviewer 2 Report

The authors have revised the manuscript significantly. No doubt, it became much better. At the same time, the new text added by the authors introduces new typos. I recommend this paper for publication after minor revision according to the comments listed below.

  1. All over the text: MMC’s and MMCs’ -> MMCs
  2. Line 66, 394: “in literature” -> “in the literature”
  3. Line 92: “with particle size finer than 3” -> “with particle size less than 3”
  4. Line 110, 386, 388: “minutes” -> “min”
  5. Line 121: “loading rate of 0.05 N/.” - ?
  6. Line 205: “present MMC’s.” -> “the present MMCs.”
  7. Lines 205, 206: “showing that O and C contamination occurs due” -> “showing that O and C contaminations occur due”
  8. Lines 271, 272: “This is explained given the local temperature increment” – not clear
  9. Line 331: there is a missing comma after TiB2.
  10. Line 346: “AT-Cu is shows the typical…” -> “AT-Cu shows the typical…”
  11. Line 353: “On the other hand, MMC-80’ the lower ductility…” -> “On the other hand, the lower ductility of MMC-80’…”

Author Response

Response to reviewer 2

Dear reviewer, thank you very much for the suggested revisions.

All over the text: MMC’s and MMCs’ -> MMCs

Corrected

Line 66, 394: “in literature” -> “in the literature”

Corrected

Line 92: “with particle size finer than 3” -> “with particle size less than 3”

Corrected.

Line 110, 386, 388: “minutes” -> “min”

“minutes” was replaced by “min”

Line 121: “loading rate of 0.05 N/.” - ?

0.05 N/s

Line 205: “present MMC’s.” -> “the present MMCs.”

Corrected

Lines 205, 206: “showing that O and C contamination occurs due” -> “showing that O and C contaminations occur due”

Corrected

Lines 271, 272: “This is explained given the local temperature increment” – not clear

The sentence has been reformulated

These white areas are portions of the Cu matrix undergoing re-melting or re-crystallization during sintering [37,46], due to the local temperature increase induced by the Joule effect [4,43].

Line 331: there is a missing comma after TiB2.

The comma has been added.

Line 346: “AT-Cu is shows the typical…” -> “AT-Cu shows the typical…”

Corrected.

Line 353: “On the other hand, MMC-80’ the lower ductility…” -> “On the other hand, the lower ductility of MMC-80’…”

Corrected.

Reviewer 3 Report

It's Ok! Now your manuscript looks cery attractive.

Good luck!

Author Response

According to reviewer 3, the paper is OK after the first round revisions.

Thank you very much for your precious suggestions.